# An I for an A: Dynamic Regulation of Adenosine Deamination-Mediated RNA Editing

**DOI:** 10.3390/genes12071026

**Published:** 2021-07-01

**Authors:** Cornelia Vesely, Michael F. Jantsch

**Affiliations:** Division of Cell & Developmental Biology, Center for Anatomy and Cell Biology, Medical University of Vienna, Schwarzspanierstrasse 17, A-1090 Vienna, Austria; Cornelia.Vesely@meduniwien.ac.at

**Keywords:** RNA modification, ADAR, regulation, RNA processing, adenosine deamination

## Abstract

RNA-editing by adenosine deaminases acting on RNA (ADARs) converts adenosines to inosines in structured RNAs. Inosines are read as guanosines by most cellular machineries. A to I editing has two major functions: first, marking endogenous RNAs as “self”, therefore helping the innate immune system to distinguish repeat- and endogenous retrovirus-derived RNAs from invading pathogenic RNAs; and second, recoding the information of the coding RNAs, leading to the translation of proteins that differ from their genomically encoded versions. It is obvious that these two important biological functions of ADARs will differ during development, in different tissues, upon altered physiological conditions or after exposure to pathogens. Indeed, different levels of ADAR-mediated editing have been observed in different tissues, as a response to altered physiology or upon pathogen exposure. In this review, we describe the dynamics of A to I editing and summarize the known and likely mechanisms that will lead to global but also substrate-specific regulation of A to I editing.

## 1. Background ADARs

Adenosine deaminases acting on RNA (ADARs) convert adenosines to inosines in structured and double-stranded RNAs (Figure 1). ADAR activity has been detected in all metazoa tested so far and genomic sequencing has revealed ADAR-like enzymes in all tested metazoan species [1].

Evolutionary, ADARs are related to Adenosine Deaminases Acting on tRNAs (ADATs), which are tRNA deaminating enzymes that are responsible for the introduction of inosines at the wobble position of certain tRNAs [2]. ADARs acquired double-stranded RNA-binding domains that allow them to bind to their substrate RNAs [3]. Depending on the organismic group, different types of ADARs exist that all harbor the deaminase domain at their C-terminal ends but contain a variable number of RNA-binding motifs (Figure 1).

In mammals, three ADAR family members can be found. The most obvious differences between these three family members are seen in the number of double-stranded RNA-binding domains and in the amino-terminal extensions of the ADAR enzymes. While ADAR1 harbors three dsRBDs, ADAR2 and ADAR3 contain only two dsRBDs. In the amino-terminal regions of the ADARs, additional nucleic acid-binding domains but also regions controlling intracellular localization of the harboring enzymes can be found (Figure 1).

## 2. Function of ADARs

In mammals, catalytic activity for ADAR1 and ADAR2 can be detected, while ADAR3 is seemingly inactive [4]. Consequently, it has been proposed that the biological function of ADAR3 lies in its ability to compete with other ADARs for RNA binding. Consistent with a function in modulating the extent of the RNA editing, loss of ADAR3 is associated with minor learning alterations in mice [5].

Loss of ADAR1 or ADAR2, in contrast, leads to dramatic phenotypes. Loss of ADAR1 leads to embryonic lethality in mice that is accompanied by liver disintegration and a dramatic increase in interferon signaling [6,7]. This phenotype can be rescued by the deletion of cytoplasmic RNA sensors Mda5 or MAVS, strongly suggesting that in the absence of ADAR1-mediated RNA editing, endogenous RNAs are interpreted as foreign or viral by the innate antiviral immune system [8,9,10]. Interestingly, ADAR1 can also prevent inadvertent activation of PKR-mediated shut-down of translation, suggesting that ADAR1 is a general modulator of RNA-sensing pattern recognition receptors [11]. Interestingly, ADAR1 is expressed in two versions. A constitutively expressed isoform gives rise to a 110 kDa protein that is mostly localized to the nucleus [12]. In contrast, an interferon-induced isoform of 150 kDa is primarily localized to the cytoplasm due to the presence of an amino terminal nuclear export signal that is missing in the shorter isoform [13,14]. The extended amino terminus of the ADAR1 p150 isoform also harbors two Z-DNA binding domains (ZBDs) that can bind left-handed Z-DNA but also Z-RNA [15]. The ZBDs have also been implicated in recognizing viral RNAs. 

The precise function of the constitutively expressed ADAR1 p110 isoform that is predominantly localized to the nucleus is not well understood. Seemingly, ADAR1 p110 cannot compensate for loss of the ADAR1 p150 as the selective deletion of the ADAR1 p150 isoform still leads to embryonic lethality [10,16].

ADAR2 is most strongly expressed in the nervous system but also in some organs such as the intestine and the vasculature [17,18]. Loss of ADAR2 in mice leads to lethality around three weeks post-partum, which is accompanied by epileptic seizures [19]. Interestingly, a knock-in of a pre-edited version of the glutamate receptor subunit 2 Gria2 can rescue this lethality, indicating that Gria2 is a major substrate of ADAR2 [20]. Still, ADAR2 also edits many other RNAs in their non-coding regions but also in their coding regions, which frequently leads to protein recoding. These protein recoding events alter protein function and therefore affect cellular and organismic physiology [21,22].

## 3. Regulation of Editing Patterns

As indicated above, editing can affect the fate and function of RNAs by several means. On the one hand, the A to I exchange introduced by ADARs can affect RNA folding and the binding landscape of proteins interacting with substrate RNAs [23]. In turn, the proteins bound to RNAs will affect the processing, localization and stability of the RNAs. Similarly, editing-induced RNA-recoding events change the amino acid sequence and therefore the function of the encoded proteins [22]. Thus, ADAR-mediated RNA editing can alter RNAs by many different means and lead to physiological changes at the cellular and organismic level [24]. 

Therefore, like many other modifications of nucleic acids, proteins, lipids or carbohydrates that affect the biochemical activities and cellular functions of these biomolecules, also ADAR-mediated RNA editing needs to be tightly controlled and regulated to assure cellular and organismic homeostasis. Indeed, it was already shown that protein-recoding RNA-editing events vary largely between tissues and throughout development [18,25]. For instance, in mammals, editing patterns are very low during embryonic development but increase rapidly post-partum [25]. Moreover, some targets show high editing patterns in a tissue-specific manner. A good example are the protein-recoding editing sites in Filamin A and Filamin B. While Filamin A editing is high in the dorsal aorta and distal colon, Filamin B editing is high in cartilage, bone and brown fat [26,27]. Similarly, Gria2 editing is regulated by neuronal activity in the hippocampus. Interestingly, this regulation is only found in the CA1 region, but not in the CA3 region of the hippocampus [28,29].

In invertebrates, such as *Drosophila* or cephalopods, the editing levels have been found to correlate with different temperatures. In *Drosophila*, this change may result from temperature-induced alterations of RNA structures [30]. In cephalopods, a large number of editing events leads to protein recoding [31]. Most interestingly, alterations in protein-recoding editing events can help to adapt protein functions to changed temperature conditions, as shown for the octopus potassium channel Kv1.1 [32].

In this review, we will focus on the different mechanisms that can affect ADAR-mediated editing. 

## 4. Genomic Variation of Substrates

As mentioned above, the double-stranded structures required for RNA editing can show temperature-dependent variability in the case of poikilotherms. Consequently, minor changes in the underlying sequences can equally affect the stability of these double-stranded structures. It is therefore not too surprising that single nucleotide polymorphisms (SNPs) have been detected in variations of *Drosophila melanogaster* that correlate with the temperatures in which these flies live [33]. In fact, these SNP variations co-occur in the surroundings of conserved A to I editing sites that show a strong temperature-dependent correlation, indicating that genomic variations can have a major impact on the extent of RNA editing. 

At a more global level, the abundance and conservation of genomic repeats is a major regulator of RNA editing. As inverted repeats within transcripts are major sites of recruitment for ADARs, the repeat repertoire in transcripts not only controls the editing levels of the repeats but also of the adjacent sites [34,35,36].

## 5. Regulation of ADAR Activity 

### 5.1. Protein Modification

Post-translational modifications (PTMs) are known to dynamically modify protein fate and function in eu- and prokaryotes. ADAR proteins are also subject to PTMs that impact on their enzymatic activity and stability.

One of the first post-translational modifications found on ADAR proteins was the SUMOylation (small ubiquitin-like modifier) of ADAR1 [37]. SUMO-1 modifies ADAR1 at lysine 418 without affecting its subcellular localization. Substitution of ADAR1 K418 by an arginine increases the editing activity of the enzyme in an in vivo reporter assay, suggesting that SUMOylation reduces editing efficiency. As K418 is located within the potential dimerization domain of ADAR1, one plausible explanation for this might be that SUMO sterically interferes with dimerization. ADAR1 and SUMO colocalize within nucleoli. This could serve as a mechanism to sequester active ADAR1 away from the nucleoplasm in order to avoid aberrant editing.

Since ADAR1 plays an important role in the attenuation of IFN induction by editing and thereby “masking” self-dsRNAs from recognition by dsRNA sensors, Li and colleagues [38] postulated that high levels of ubiquitously expressed ADAR1p110 might be inhibiting any IFN-dependent antiviral defense mechanism of the cell. Indeed, upon IFN stimulation, ADAR1p110 is polyubiquitinated on lysine 48 by E3 ligase b-transducin repeat-containing protein (b-TrCP) and degraded. Two additional lysins (K574 and K576) are essential for this ubiquitination. More importantly, inhibition of the ubiquitination-dependent ADAR1-p110 downregulation limits IFN-mediated antiviral activity in HEK293T cells.

Both active ADARs have been identified as targets of protein phosphorylation mainly in the context of high-throughput phosphoproteomic studies (PhosphoSite). However, to date only few sites have been experimentally proven to affect enzymatic activity of the editases. More recently, the editing activity of ADAR1p110 and ADAR2 was found to be regulated by phosphorylation of specific sites [39]. One site in each enzyme is phosphorylated by nuclear thymoma viral oncogene homolog (AKT) kinases (AKT-1, -2 and -3). ADAR1p110 phosphorylation of T738 and ADAR2 phosphorylation on T553 results in a 50–100% reduction in ADAR-dependent editing efficiency on selected editing targets. Since the phosphorylation happens directly within the deaminase domain of the two enzymes, it could interfere with the charge distribution in the active center, making a nucleophilic attack impossible or representing a steric blockade. Thus, by changing the levels of AKT kinases and/or actively removing the phosphorylation by an unknown phosphatase, the editing activity of the ADAR proteins can be regulated (Figure 2).

### 5.2. Turnover and Cellular Localization

ADAR localization and stability can also be used to regulate editing efficiency in a cell. Some factors that influence either localization or protein stability have been identified to date. ADAR2 is regulated by binding to the prolyl-isomerase PIN1. Binding requires a phosphorylated recognition motif (P-Ser/P-Thr-Pro) in the N-terminal region of ADAR2. Upon binding, PIN1 catalyzes the cis/trans isomerization of the peptide bond, resulting in a conformational change that positively affects the ADAR2 editing activity. In the absence of PIN1, ADAR2 is mis-localized to the cytoplasm and marked for degradation by ubiquitination. The responsible enzyme, the E3 ubiquitin ligase WWP2, binds to two conserved PPxY motifs within the N- and C-terminus of ADAR2 and consequently poly-ubiquitinates ADAR2. This, in turn, leads to the degradation of ADAR2 by the proteasome, and thus a reduction in the ADAR2 protein levels. [40]. The ADAR1 and ADAR2 protein levels can also be regulated by another protein. AIMP2, a component of the aminoacyl-tRNA synthetase complex that seems to have a non-canonical function in protein stability regulation, interacts with both active ADARs, and leads to a reduction in their protein levels. In skeletal muscle, where AIMP2 is highly expressed, this for the biggest part explains the overall low editing found in muscle [18].

Regulation of editing is often required in dynamic processes such as development, tumorigenesis or temperature adaptation. One example for this is the regulation of the nuclear ADAR2 protein levels by importin-α4 during the maturation of neurons. As they mature, more importin-α4 is expressed and interacts with ADAR2, thereby fostering its nuclear localization. Within the nucleus, ADAR2 interacts gradually more with the isomerase PIN1. This way, despite the fact that the protein levels of the ADARs are constant during brain development, the ADAR2 editing efficiency is higher in mature neurons compared to immature ones [41].

Similarly, ADAR1 seems to be regulated during regeneration, as demonstrated by a study in newts, *Notophthalmus viridescens*. These amphibians are known for their great regenerative capacity. Upon injury, the ADAR1 protein is shuffled from the cytoplasm into the nucleus and therefore able to edit nuclear targets, while after complete regeneration ADAR1 is found in the cytoplasm again. This nucleocytoplasmic shuttling of ADAR1 ensures editing of the target RNAs during the repair process [42].

Despite the fact that the editing levels generally rise during development while the ADAR protein levels stay relatively constant, editing regulation is also found at the level of transcription. However, only in certain tissues or cell types. For instance, in hippocampal neurons, the transcription factor CREB (cAMP response element-binding protein) induces expression of ADAR2, thereby protecting them from forebrain ischemia [43]. Similarly, ADAR1 expression is regulated by TARDBP in HepG2 cells. TARDBP encodes the TDP-43 protein that is able to bind DNA and RNA and regulates transcription and RNA processing. By binding to several regulatory elements in the Adar gene, TDP-43 enhances ADAR1 expression [44].

Human ADAR1 harbors a nuclear localization signal that is composed of a bipartite motif that flanks the third double stranded RNA-binding domain [45]. Interestingly, RNA-binding can mask this nuclear localization signal [46]. Bound double-stranded RNA may also be recognized by exportin 5. Consequently, ADAR1 p110 and p150 can shuttle between the nucleus and cytoplasm, where the RNA binding-status can affect their intracellular localization [45]. Still, ADAR1 p150 is mostly found in the nucleus, while the p110 version is predominantly nuclear [45].

ADAR2 harbors an N-terminal nuclear localization signal and is exclusively localized to the nucleus [47]. Similarly, ADAR3 is also localized to the nucleus [4]. Within the nucleus, ADARs can be found enriched within nucleoli [47,48]. To this point, no A to I editing of ribosomal RNAs has been detected. However, editing of miRNAs has been detected [1,49,50,51,52]. It is therefore possible that the nucleolar localization of the ADARs reflects their activities on these small RNAs. An alternative explanation for the localization of ADARs in nucleoli may be their sequestration to this compartment. This may aid in controlling their intracellular levels. Indeed, overexpression of substrate RNAs for ADAR2 have shown to lead to increased release of ADAR2 from nucleoli [53]. Moreover, SUMOylation has been shown to affect the nucleolar concentration of ADAR1 [47].

Thus, the localization of ADARs may modulate enzymatic activity. The NES-bearing cytoplasmic ADAR1-p150 isoform is expressed from an interferon-regulated promoter [54]. Inflammation-mediated interferon signaling thus leads to the cytoplasmic accumulation of this isoform and therefore will boost editing of cytoplasmically localized RNAs [55,56]. In contrast, the nucleo-cytoplasmic shuttling of ADAR1 p110 is controlled by its RNA-bound state [45]. It is not clear yet whether a specific class of RNAs can control p110 localization. It is tempting to speculate, however, that the same targets of ADAR1 p110 will control its intracellular localization and therefore its interaction with other substrates. Similarly, ADAR2 has been shown to shuttle in and out of the nucleolus depending on the presence of nuclear-editing substrates [53]. Thus, also here the availability of certain substrate RNAs may affect the localization and substrate interaction of ADAR2 (Figure 2).

### 5.3. RNA Processing Dynamics

Editing occurs predominantly in the nucleus and happens co-transcriptionally. The very same is true for (alternative) splicing. All are dynamic processes that not only expand the diversity of the transcriptome but also influence each other. Alternative splicing can regulate editing by producing different ADAR isoforms, which enhance, diminish or abolish editing activity [57,58,59,60,61,62].

Especially for the ADAR2 transcript, a number of alternative splicing options have been documented. While not all of them are well conserved, most of them influence either the ADAR protein levels by introducing premature stop codons (PTC) or they affect editing efficiency directly. The inclusion of an alternative exon comprising an Alu (exon 5a) reduces the editing efficiency of ADAR2. Skipping of exon 2, which contains two of the dsRNA binding domains, results in a frameshift that introduces a premature stop. Further, by editing its own mRNA, ADAR2 can autoregulate itself. Self-editing creates an alternative 3´splice site that leads to an insertion close to the translational start site and introduces a premature stop codon [63,64].

The splicing patterns of ADAR1, apart from its different promoters for interferon-inducible p150 and constitutively expressed p110, are less complex than for ADAR2. One study reported an alternative intron within exon 2 of ADAR1, which is either part of the 5´UTR for ADAR1p110 or within the ORF for ADAR1p150 and leads to a PTC. However, this short transcript seems to be left untouched by nonsense-mediated decay [61].

Since RNA editing and splicing basically appear at the same time and location, it can be expected that these two processes have extensive crosstalk. Thus, it is not surprising that splicing is able to regulate editing. This regulation depends on the efficiency of the splicing process. For editing sites that are defined at least in part by intronic regions, the editing levels are reduced, if the intron is spliced out quickly [65]. On a genome-wide level, splicing inhibition therefore leads to an increase in the editing levels at several hundred sites. Moreover, the lack of two alternative splicing factors, NOVA1 and NOVA2, is able to perturb the overall editing landscape [66].

Very recently, another more direct mode of regulation of ADAR-editing efficiency was demonstrated, which is brought upon by a change in intracellular pH. The authors could show that intracellular acidification modulates ADAR catalytic activity directly. This is possible due to enhanced flipping out of the adenosine to be edited and additionally an increased deamination rate at a more acidic pH [67] (Figure 2).

### 5.4. Competitive RNA-Binding

One of the *trans* regulators or competitors of active ADAR proteins comes from within the family. ADAR3, although not an active editase, has a high sequence similarity to ADAR1 and ADAR2 and additionally contains a unique R-domain that allows it to bind ssRNA. ADAR3 is exclusively expressed in the brain, and there it is negatively correlated with overall editing levels [18]. In vitro studies show reduced editing of the 5-HT2cR RNA by ADAR1 and ADAR2 due to ADAR3 binding [4]. ADAR3 is also upregulated in glioblastomas, inhibiting editing of the GluR-B receptor (*Gria2*) mRNA in vivo. In this example, ADAR3 binding to *Gria2* mRNA also leads to the inhibition of editing [68].

In a cell with hundreds of different RNA-binding proteins, the heat is on each single RBP to bind to its target RNAs without being outcompeted by other RBPs that can bind as well. This very much depends of course on the spatial and temporal abundance of certain competing proteins. In that sense, most editing regulators identified were specific to only a set of target sites. These site-specific regulators include Fragile X proteins, FMRP and FXR1P, in *Drosophila* and human brain [69,70], and *Dm* Maleless and its human homolog DHX9, in splicing coordination and cancer, respectively [71,72]. 

Other RBPs with clear site-specific regulation of editing are RPS14, SFRS9 and DDX15 regulating ADAR2 targets [73], and Ro60, which regulates editing of a set of ADAR1 targets [44]. Further known interactors of ADARs that impact more broadly on editing activity are the nuclear factor 90 (NF90) proteins, NF90/ILF3 [74] and NF45/ILF2 [75], and important proteins of the microRNA processing pathway, DROSHA and DGCR8 [44,76]. While NF90 proteins interact in an RNA-dependent fashion, DROSHA only forms protein–protein interactions and thereby positively regulates ADAR editing. How this editing enhancement by DROSHA is achieved remains to be demonstrated. NF90 proteins, ILF2 and especially ILF3 are broadly influential negative regulators of editing, as demonstrated by knockdown and overexpression studies [44,77].

More recently, two additional interactors of ADAR proteins were identified that regulate editing by RNA binding: The first, a zinc finger RBP, Zn72D, was identified as a regulator of ADAR in the fly brain. Zn72D interacts with ADAR in an RNA-dependent manner and enhances editing of roughly two thirds of the tested editing sites. Its ADAR regulatory function is conserved as the mouse homolog, Zfr, also interacts with and regulates ADAR2 and, to a lesser extent, ADAR1 in mouse primary neurons [77,78]. One structural feature that ILF3, ILF2 and ZFR have in common is a so-called DZF domain. This domain seems to facilitate protein dimerization and RNA binding [79] and is also found in the other protein identified as a new negative ADAR regulator in the same study, STRBP, which is a paralog of ILF3 [77].

However, the exact binding sites of these RBPs and thus their mode of action is still missing. Additionally, one has to keep in mind that effects of most of these RBPs on editing greatly depend on the cell or tissue type. This is nicely demonstrated, among others, by the study of Quinones-Valdez and colleagues that looked at RBP-KDs in two different cell lines (K562 and HepG2), resulting in two different and only partially overlapping sets of ADAR1 regulators.

### 5.5. Physiological Stimuli

The dynamic regulation of RNA editing strongly suggests that A to I deamination can respond to environmental or physiological changes. Such changes in the surroundings of cells and organisms will obviously need to be translated into biochemical signals, which have been outlined above. Still, at this point, the pathways that relay extracellular signals into editing patterns are not understood. To this end, signals that can affect editing include temperature [30], social stress [80] or neuronal activity [29,81]. It is likely that these examples represent the tip of the iceberg and that many other mechanisms controlling editing activity are yet to be discovered (Figure 2). 

## 6. Outlook

Understanding editing regulation is important for therapeutic approaches. Today, ADARs are exploited as tools to repair and alter the coding information of RNAs for therapeutic means. To do so, ADAR-derived constructs can be introduced to cells that are targeted to specific substrates. However, another approach is the recruitment of endogenous ADARs to specific target transcripts by artificially generating double-stranded structures using antisense oligonucleotides. Here, it is still not clear which ADAR isoform is best suited to be recruited to the site of editing. Obviously, the enzyme to be recruited needs to be expressed in the respective tissue or cell type. Since the expression of ADAR2 is relatively restricted, it seems that ADAR1 and its isoforms may be more suitable to direct site-directed editing. Still, the different intracellular localizations and modifications of the proteins may be important factors to be considered as the antisense RNA will need to be delivered to the same cellular component. 

## Figures and Tables

**Figure 1 genes-12-01026-f001:**
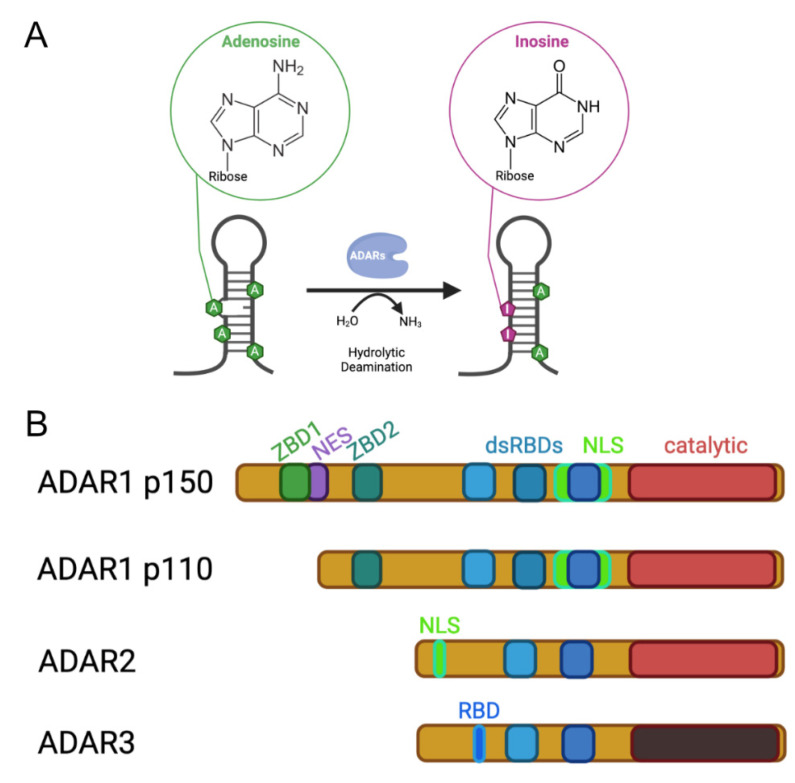
Activity and architecture of mammalian ADARs. (**A**) Adenosine deaminases acting on RNA convert adenosines via hydrolytic deamination to inosines. (**B**) Three ADAR genes can be found in mammals, ADAR1, ADAR2 and ADAR3. ADAR1 is expressed in two isoforms, ADAR1 p150 of 150 kDa, which is expressed from an interferon-inducible promoter, and ADAR1 p110, which is constitutively expressed. All ADARs contain a deaminase domain at their C-terminus. However, the catalytic domain of ADAR3 is enzymatically inactive. Substrate engagement of the ADARs is mediated via a variable number of double-stranded RNA-binding domains (dsRBDs) that interact with structured and double-stranded RNAs. The amino termini of the ADAR proteins are quite divergent. While the N-terminus of ADAR1 harbors a Z-DNA binding domain (ZBD) that is able to interact with Z-DNA and ZRNA, the long N terminus of ADAR1 p150 also harbors a nuclear export signal (NES), resulting in mostly cytoplasmic localization of this interferon-induced isoform. A nuclear localization signal (NLS) that overlaps the third dsRBD of ADAR1 is negatively regulated by RNA binding. As a consequence, the p150 isoform of ADAR1 can shuttle between the nucleus and cytoplasm. The N-terminus of ADAR2 harbors its classical nuclear localization signal. The catalytically inactive ADAR3 protein contains an N-terminally located RNA-binding domain (RBD).

**Figure 2 genes-12-01026-f002:**
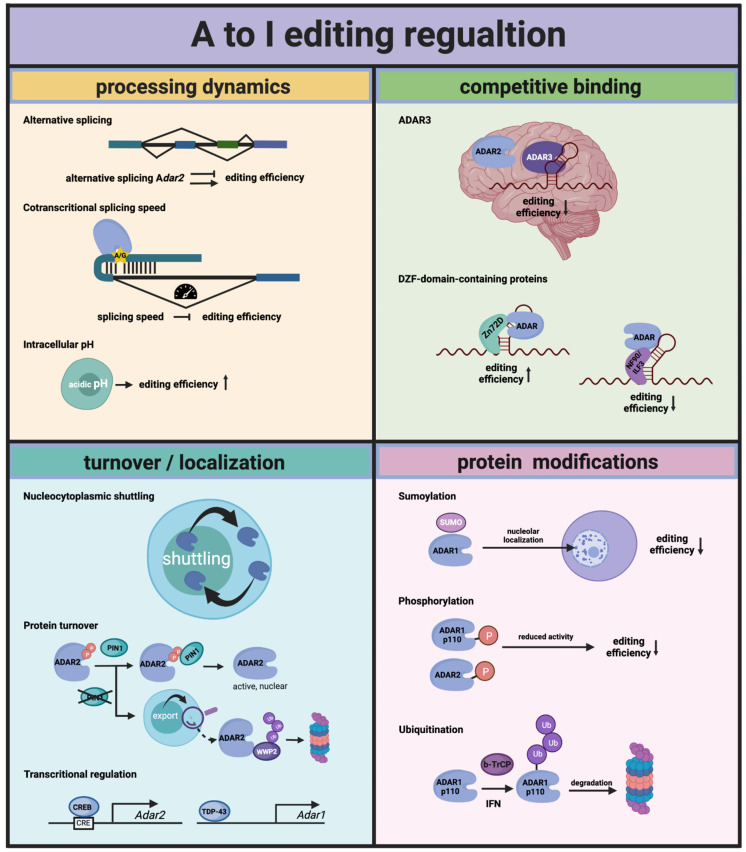
Regulation of A to I editing by Adenosine Deaminases Acting on RNA (ADARs). *Adar* transcripts can be alternatively spliced, resulting in premature termination codons or the expression of an ADAR protein with modified editing activity. The time for ADARs to edit sites that consist of secondary structures in which intronic sequences are contained is restricted by the speed of the splicing that removes the intron. Intracellular acidic pH seems to favor the flipping out of the adenosine that is edited, resulting in higher editing rates. Overall, the editing rates for different types of editing sites (intronic, exonic and UTRs) is controlled by the ability of the ADAR proteins to shuttle between the nucleus and the cytoplasm. PIN1, a prolyl-isomerase, stabilizes ADAR2 in the nucleus and thereby regulates editing of the nuclear targets of ADAR2. Further, levels of ADARs can be regulated on a transcriptional level, by CREB for ADAR2 and by TDP-43 for ADAR1. Competitive binding of other RNA-binding proteins in general is another way to regulate editing. In many cases, however, this happens in a more tissue- or target-specific way. More broadly, the influential competitors of ADARs are ADAR3 in the brain and some proteins that have a DZF domain in common, such as NF-90 and Zn72D. Finally, ADARs can be decorated by different protein modifications (SUMOylation, phosphorylation and ubiquitination), which lead to either a reduction in available editase or compromise its editing activity.

## Data Availability

No new data were created or analyzed in this study. Data sharing is not applicable to this article.

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
