# Peer review of "An I for an A: Dynamic Regulation of Adenosine Deamination-Mediated RNA Editing"

_genes, 2021, doi:10.3390/genes12071026_

Round 1

Reviewer 1 Report

The article is valuable and interesting for the broad scientific community.

The role of adenine deamination in the antivirus defense system is important, especially in the time of Covid-19.  The article is well written and readable.

However, the scheme of Inosine formation in the cell should be presented in the manuscript.

Reviewer 2 Report

In this review manuscript, the authors summarize the current status of A-to-I RNA editing. The authors nicely introduce this field in the manuscript. Thus, this reviewer has only two comments below:

Major points:

[1] “Background ADARs”. The paragraphing of this part needs much work as only one sentence forms the second paragraph. The same applies to “By turnover and cellular localization” section.

Minor points:

(1) Page 1.  ADATs must be spelled out at its first appearance.
